# READER: Retrieval-Assisted Drafter for Efficient LLM Inference

## Abstract

Autoregressive Language Models instantiate a factorized likelihood over token sequences, yet their strictly sequential decoding process imposes an intrinsic lower bound on inference latency. This bottleneck has emerged as a central obstacle to the scalable deployment of large-scale generative models. Existing acceleration techniques partially mitigate token-level latency — by relying on auxiliary draft models or introducing an additional training phase — but fail to address the dominant memory and communication costs. We present **READER** (Retrieval-Assisted Drafter for Efficient LLM Inference), a *provably lossless* speculative decoding framework that reuses an existing trained draft model as its backbone and requires no additional retraining. READER formalizes speculative decoding as a stochastic tree construction problem and exploits the empirical redundancy structure of natural language to generate high-probability candidate continuations. Our method revisits the problem of constructing draft trees, establishing substantial statistical improvements over stochastic draft-tree methods and providing a complexity-theoretic analysis that characterizes the optimality frontier of speculative decoding under bounded computation and memory resources. Beyond the single-sequence regime traditionally considered in prior work, we introduce a memory-optimal key-value cache-serving strategy that guarantees amortized sublinear overhead in the batch dimension, allowing READER to scale to realistic inference workloads. Comprehensive experiments demonstrate up to **6.13×** wall-clock speedup on single-prompt inference and up to **5.92×** on batched inference — consistently surpassing prior speculative decoding baselines — while preserving exact output equivalence, with even more pronounced gains in retrieval-augmented generation pipelines. Our results close a key gap between theoretical parallelism limits and practical LLM inference, suggesting a new standard for efficient deployment.

## 1 Introduction

The widespread adoption of large language models (LLMs) has drawn attention to their substantial energy costs (Strubell et al., 2019), motivating extensive research on improving inference efficiency. Recently, reasoning-focused models such as OpenAI o1 (Jaech et al., 2024) and DeepSeek-R1 (Guo et al., 2025) have emerged. These models achieve strong performance by generating longer "thinking" trajectories at inference time. While inference-time scaling improves accuracy, it also dramatically increases the number of generated tokens, exacerbating latency and energy costs (Zhang et al., 2025). This makes efficient decoding strategies critical for the next generation of reasoning LLMs.

LLMs generate tokens autoregressively, one at a time. This strictly sequential dependency inherently resists parallelization: each decoding step requires a full forward pass conditioned on all previously generated tokens. As model sizes and context lengths grow, the cost of this step-by-step process becomes even more demanding. In practice, memory and communication overheads dominate. Each token requires accessing and updating the Key-Value (KV) cache, whose bandwidth demands become the primary bottleneck in high-throughput or long-context settings.

A promising line of work that seeks to reduce this sequential bottleneck is speculative decoding (Stern et al., 2018; Leviathan et al., 2023). At its core, speculative decoding decouples speculation from verification: a candidate continuation is generated in parallel, and the base model verifies the

entire block in a single forward pass. If the candidate aligns with the base model's distribution, multiple tokens are accepted at once, collapsing many sequential steps into one. Early work has shown that even simple draft strategies can yield significant speedups without changing the underlying model's output distribution (Chen et al., 2023; Xia et al., 2024). Despite this, existing speculative decoding approaches leave important gaps. Speedups often remain bounded by shallow draft structures or by additional overheads, while scaling to realistic batch-serving workloads remains ineffective.

In this paper, we present READER (Retrieval-Assisted Drafter for Efficient LLM Inference), a *provably lossless* speculative decoding framework that directly addresses these challenges. READER exploits the theoretical limits of draft-tree expansion, generating high-probability continuations without requiring additional training. An important contribution of READER is its complexity-theoretic analysis of speculative decoding: we characterize the optimality frontier under bounded computation and memory, analyzing the inherent limits of draft construction strategies. Beyond single-sequence decoding, READER introduces a memory-optimal KV cache rearrangement strategy that guarantees amortized sublinear overhead in batch serving. This makes speculative decoding viable at the scales relevant for modern LLM deployment.

READER augments the standard drafting with a *heterogeneous* tree that blends tokens from two sources: (i) a lightweight speculator that proposes short, high-confidence branches, and (ii) a deterministic retrieval path constructed via CPU-side search over a short-term trie (prompt and generated history) and a long-term datastore indexed with a suffix array. We attach a *deep* retrieval-driven branch to the root ("widening") and *deepen* internal nodes by seeding search from partial draft-model prefixes. This design preserves a fixed per-sample tree shape, while substantially increasing token diversity at negligible marginal latency.

Experiments across diverse tasks demonstrate that READER achieves up to **6.13x** wall-clock speedup on single-prompt inference and up to **5.92x** on batched inference, consistently surpassing prior speculative decoding baselines, with especially pronounced gains in retrieval-augmented generation pipelines with more than **10×** speedup. By pushing speculative decoding closer to its theoretical parallelism limits, READER advances the efficiency frontier of LLM inference.

**Paper organization** This paper is organized as follows. Section 2 reviews the speculative decoding setup and the roles of the draft and base models. Section 3 provides a comprehensive theoretical analysis of speculative decoding with a heterogeneous tree structure, establishing the foundations for our method. Section 4 presents the READER algorithm along with a brief pseudocode sketch. Section 5 reports our experimental setup and results. Section 6 concludes the paper. The appendix contains full pseudocode and implementation details.

## 2 RELATED WORK

A large number of studies accelerate LLM inference through model compression. Quantization methods such as LLM.int8 (Dettmers et al., 2022), SmoothQuant (Xiao et al., 2023) and AWQ (Lin et al., 2024) reduce activation and weight precision while preserving accuracy, and pruning approaches like SparseGPT (Frantar & Alistarh, 2023) remove redundant weights with minimal perplexity increase. Knowledge distillation can further shrink models for faster decoding. These approaches, however, typically modify the model and may introduce accuracy drops (Lang et al., 2024), while our goal is lossless acceleration of the unmodified base model.

Another line of research targets the memory- and system-level bottlenecks of autoregressive decoding. FlashAttention optimizes attention with IO-aware kernels (Dao et al., 2022), multi-query attention reduces key-value storage by sharing across heads (Shazeer, 2019), and systems like vLLM introduce paged KV caches and continuous batching to improve throughput (Kwon et al., 2023). Such techniques are complementary to speculative decoding, which reduces the number of sequential steps rather than the cost of each step.

Speculative decoding (Leviathan et al., 2023) itself originates from blockwise parallel decoding (Stern et al., 2018) and speculative sampling (Chen et al., 2023). In these methods, a small *drafter* proposes multiple tokens while the base model *verifies* them in one pass, accepting the longest correct prefix to preserve exactness. Learned drafters such as Medusa (Cai et al., 2024), EAGLE

family (Li et al., 2024a;b; 2025) increase acceptance by aligning proposal distributions with the verifier and by constructing deeper, context-aware draft trees. Other draft-based approaches have explored alternative designs: CTC-style drafters that exploit conditional independence for parallel speculation (Wen et al., 2024), diffusion-based drafters that generate multi-token proposals through iterative refinement (Christopher et al., 2024), and hybrid models such as SpecInfer (Miao et al., 2024). A complementary "self-speculative" direction derives the drafter from the base model itself (e.g., layer skipping / early-exit) to avoid an auxiliary network while remaining lossless under strict verification rules (Zhang et al., 2024). Despite these gains, trained or self-derived drafters can introduce additional engineering, storage, or scheduling complexity, and their speedups often depend on careful batching and KV-cache management.

In contrast, training-free approaches avoid extra model training by leveraging explicit structure in text. REST drafts from a retrieval datastore to capture frequent local continuations, while ANPD constructs and adapts an $n$-gram module online using real-time statistics (He et al., 2024; Ou et al., 2024). Lookahead decoding dispenses with any drafter entirely by expending more compute per step to verify multiple $n$-gram candidates directly with the base model, trading FLOPs for fewer sequential steps while remaining exact (Fu et al., 2024). These strategies are attractive for their simplicity and exactness, yet they often under-exploit the statistical regularities that govern acceptance across branches or ignore serving-time constraints (e.g., KV-cache bandwidth and batch scheduling).

Finally, several recent studies emphasize batch-serving and KV-cache efficiency. MagicDec demonstrates that speculative decoding can improve both latency and throughput when caches are managed carefully (Sadhukhan et al., 2024). EAGLE-3 paper (Li et al., 2025) also provides analysis on large batch size acceleration.

## 3 THEORETICAL ANALYSIS

In this section, we present a complexity-theoretic analysis of speculative decoding with tree attention and examine the potential for theoretical acceleration in speculative decoding methods.

### 3.1 SPECULATIVE DECODING

Model-based speculative decoding employs an auxiliary draft model, also referred to as a *speculator*. In each forward pass, the draft model generates multiple candidate tokens predicting the continuation of the output sequence. These tokens are then validated in parallel by the base model within a single forward pass. If the predictions are confirmed, multiple tokens can be committed in a single inference step. The degree of alignment between the predicted tokens and the true continuation directly determines the speedup achieved.

The depth of speculation influences the cost of the drafting stage. For model-based approaches, this cost scales linearly with the depth of speculation, as it requires one call to the draft model.

As shown in Leviathan et al. (2023), using the draft token acceptance probability $\alpha$, the expected acceptance length for a single-branch draft sequence of length $\gamma$ is

$$E(\gamma, \alpha) = \mathbb{E}\left[\text{acceptance length}\right] = \frac{1 - \alpha^{\gamma+1}}{1 - \alpha}. \tag{1}$$

Sadhukhan et al. (2024) extend this analysis to batched inference. Let $T_V(\gamma, B, S)$ denote the time required for the base model to verify $\gamma$ draft tokens with batch size $B$ and KV-cache size $S$, and let $T_D(B, S)$ denote the time to generate one draft token under the same conditions. Then, the time for a single decoding step is

$$T_{SD}(\gamma, B, S) = \gamma \cdot T_D(B, S) + T_V(\gamma, B, S). \tag{2}$$

The corresponding average acceleration relative to autoregressive decoding is

$$E(\gamma, \alpha) \cdot \frac{T_{AR}}{T_{SD}(\gamma, B, S)} = \frac{1 - \alpha^{\gamma+1}}{1 - \alpha} \cdot \frac{T_{AR}}{\gamma \cdot T_D(B, S) + T_V(\gamma, B, S)}.$$

Optimizing over $\gamma$ yields the optimal number of draft tokens. This formulation, however, applies only to single-branch decoding with a draft model. Our analysis generalizes this framework to (1)

tree-structured decoding and (2) heterogeneous draft tokens obtained from multiple sources. In the following sections, we formalize heterogeneous tree-structured speculative decoding and establish a theorem on the optimal tree structure for acceleration.

## 3.2 TREE DECODING

Tree decoding (Miao et al., 2024; Sun et al., 2023) augments model-based speculative decoding by expanding multiple plausible continuations per step. The speculator proposes several high-probability tokens at each node, forming a decoding tree; the base model then verifies all proposed tokens in parallel. This increases the expected acceptance length and mitigates memory stalls by processing the tree jointly. We also consider decoding mask $M \in \{0, 1\}^{\gamma \times \gamma}$, with $M_{ij} = 1$ if and only if node $i$ is an ancestor of node $j$, which is consistent across the batch. During verification, this mask restricts each query to attend only to its ancestral path in the tree, ensuring that attention scores are computed only along valid draft prefixes.

Let $b$ be the batch size, $\gamma$ the tree size (speculative length), $s$ the KV-cache length, and $h$ the hidden dimension. We measure cost in FLOPs and memory reads.

Computing $Q, K, V$ for the $\gamma$ new tokens (and the output $Y$) requires

$$\Theta(b\gamma h^2) \text{ FLOPs}, \qquad \Theta(b\gamma h + h^2) \text{ reads}.$$

Scaled dot-product attention over the $\gamma$ queries against $s + \gamma$ keys/values requires

$$\Theta(bh\gamma(\gamma + s)) \text{ FLOPs}, \qquad \Theta(bh(\gamma + s)) \text{ reads}.$$

The FLOPs/read ratio is $\Theta(h) = O(1)$ for $Q, K, V$ and $Y$, hence these stages remain memory-bound (for fixed $h$). For attention it is $\Theta(\gamma)$, so as $\gamma$ grows the computation dominates. Consequently, increasing $\gamma$ is effectively free while inference is memory-bound; beyond the compute-bound regime, $\gamma$ should be increased only if the expected acceptance length grows faster than the attention compute, i.e., faster than $\Theta(\gamma)$.

## 3.3 OPTIMAL TREE-STRUCTURED HETEROGENEOUS SPECULATIVE DECODING

Tree-structured drafting is widely used in speculative decoding but tends to push verification into the compute-bound regime; consequently, adding low-acceptance tokens can reduce throughput. We analyze *heterogeneous* trees in which tokens may be proposed by different mechanisms (model-based, self-speculative, search-based), each with its own generation cost.

Fix a rooted tree $T$ with $|T|$ nodes. For node $i$ at depth $d(i)$, let

$$\alpha_i = \Pr\left(X_{1:d(i)} = \text{prefix}(i)\right)$$

be its *acceptance frequency* (cumulative prefix probability). Equivalently, $\alpha_i$ is the limiting empirical frequency with which node $i$ is accepted if verification is repeated over i.i.d. draws. For simplicity of the following derivation, we also use $\alpha_0 = 1$ is the acceptance rate of the root token (which is generated by the base model). These rates depend on the tree structure $T$, dataset and base model's output distribution.

**Lemma 1** (Tree-structured expected accepted length). *The expected number of accepted* speculative *tokens when verifying against $T$ is*

$$E(T) = \sum_{i=0}^{|T|} \alpha_i.$$

This expression does not depend on node depths beyond their role in determining $\alpha_i$. When $T$ is a single branch of length $\gamma$, Lemma 1 reduces to the single-branch formula (Equation (1)), up to whether the depth-0 root is counted in the acceptance length.

Different drafting mechanisms incur different costs. We model these via per-node *generation times*. In practice, one of the following obtains:

1. *Layerwise:* a single forward pass generates an entire depth layer $l$ (cost $t_1^l$);
2. *Treewise:* a single forward pass generates the entire tree (cost $t_2$);

3. *Nodewise:* each node is generated individually (cost $t_3^i$ for node $i$).

To unify these cases, we define an *effective* per-node time $t_i \geq 0$ by apportioning layerwise or treewise costs to nodes (e.g., divide $t_1^l$ equally among nodes in layer $l$, and divide $t_2$ equally among all nodes). This yields the following lemma.

**Lemma 2** (Drafting time). *With effective per-node times $\{t_i\}_{i \in T}$, the total drafting time is*

$$T_D(T) \;=\; \sum_{i=1}^{|T|} t_i.$$

Formal justification of this reduction, along with illustrative scheme, is provided in Section C.2.

Let $T_V(T)$ denote the verification time for tree $T$. While $T_V(T)$ scales linearly (see Section 3.2) with $|T|$ under fixed architecture, its constants are hardware dependent. Combining Lemmas 1 and 2 we can formulate the following theorem:

**Theorem 1.** *An optimal tree structure solves*

$$T^* \in \underset{T - rooted\ tree}{\arg\max} \; \left(\sum_{i=0}^{|T|} \alpha_i\right) \cdot \left(\sum_{i=1}^{|T|} t_i + T_V(T)\right)^{-1}. \tag{3}$$

Proof is deferred to Section C.3. This objective immediately implies that nodes with large ratios $t_i/\alpha_i$ harm acceleration and should be pruned, subject to the interaction captured by $T_V(T)$. In Section F we provide a simple constructive algorithm for tree selection based on this criterion. When a closed-form or empirically calibrated model for $T_V(T)$ is available, the optimization can be evaluated accurately for a given hardware stack.

## 4 READER

In this section, we introduce READER, a speculative decoding algorithm, based on the heterogeneous draft tree, which has optimal structure and optimized KV Cache serving strategy. Firstly, we provide the theoretical analysis of the draft model predictive ability and search-based theoretical upper-bounds.

### 4.1 ACCEPTANCE LENGTH OF HETEROGENEOUS TOKENS

We study the average acceptance length—the number of consecutive draft tokens accepted per forward pass—in model-based speculative decoding. We also introduce a *self-repetitiveness* metric for natural text that provides a theoretical upper bound on the acceptance length achievable by search-derived tokens.

We begin by measuring acceptance statistics for the draft model on GSM (mathematics) (Cobbe et al., 2021) and HumanEval (coding) (Chen et al., 2021). Figure 1 reports the distribution of accepted tokens per forward pass for Llama-3.1-8B-Instruct (Grattafiori et al., 2024) using the EAGLE-3 speculative method. We set the draft tree depth to 8 and the total number of draft tokens to 60, with batch size 1 on an 8B LLM. In approximately $30\%$ of forward passes, the verifier accepts the maximum number of draft tokens.

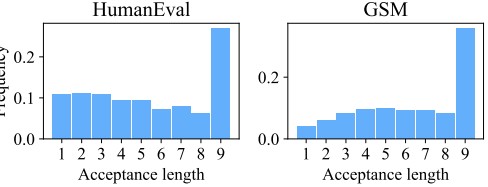

Figure 1: Acceptance length distribution for HumanEval (left) and GSM (right) datasets

| Dataset | W/o DS | W/ DS |
|---|---|---|
| Magpie-Qwen2.5-Coder | 1.38 | 1.90 |
| Magpie-R1-Llama-70B | 2.86 | 3.54 |
| Hagrid (RAG) | 10.32 | — |

Table 1: Self-repetition metric for different datasets (W/o DS: without datastore; W/ DS: with datastore, consisting of 100 responses that are not used for metric calculation. This datastore can be built online by appending generated responses)

READER's draft tree also includes *search-based* tokens. Because these tokens exploit repetitions in the target text, more repetition leads to faster inference. We are particularly interested in *long* repetitions: short repeats are typically captured by the draft model (as shown above), whereas long repeats are where search contributes most. This phenomenon appears in both human-written and model-generated natural text, and it is especially relevant for code generation—one of the most important LLM applications.

To quantify how repetitions accelerate inference, we define the following metric. Given a tokenized input (prompt) and a pre-generated response, apply:

1. Place a pointer at the start of the response.
2. Find the longest substring beginning at the pointer that also appears in the input or in the already-processed prefix of the response.
3. If no continuation is found (length zero), advance the pointer by one token; otherwise, advance it by the continuation length.

The metric equals the total number of response tokens divided by the number of pointer advances. This value matches the average acceptance length of speculative decoding with an *infinite* decoding tree that indexes all previously seen history. In practice, inference can further benefit from a large external datastore of tokens drawn from other prompts, which speeds up common phrases.

Table 1 reports this metric on several datasets with generated answers—coding, chain-of-thought (Xu et al., 2024), and RAG (Kamalloo et al., 2023). The results indicate that for tasks such as reasoning and RAG, where repetitions are frequent, our approach can substantially accelerate existing speculative decoding methods.

## 4.2 ALGORITHM

Our approach accelerates speculative decoding by constructing a *heterogeneous draft tree* that blends search- (retrieval-) derived tokens with tokens proposed by a draft model. Building on the theoretical foundations, we empirically identify effective tree shapes and techniques that raise the quality of drafted tokens.

For the *short-term context*, we maintain a trie that supports:

1. inserting a sequence $S$ in $O(|S|)$ time;
2. searching for a sequence $S$ in $O(|S|)$ time.

During decoding, the trie is populated with the input prompt, self-generated tokens, and (optionally) external text. To build the draft tree, we take a suffix $S$ of the generated tokens, descend the trie with $S$, and extract a subtree of a prescribed shape. The suffix length is a hyperparameter. If $S$ is not present, we drop its first token and retry.

A common extraction strategy fixes both a maximum depth and a maximum token budget, then performs a depth-first traversal subject to these limits. To improve acceptance length, trie nodes are sorted by continuation frequency so that high-frequency successors are explored first.

To capture *long-term context*, we augment the system with a large auxiliary datastore comprising many responses, ideally produced by the base model. A trie is impractical here due to memory growth at scale. Because this datastore is static at inference time, we index it with a *suffix array*. Lookups reduce to binary search over prefixes since the array stores substrings in lexicographic order. To ensure broad task coverage, the datastore mixes texts from diverse sources; we use the Magpie dataset in our experiments. The resulting index fits in RAM and remains under 1 GB.

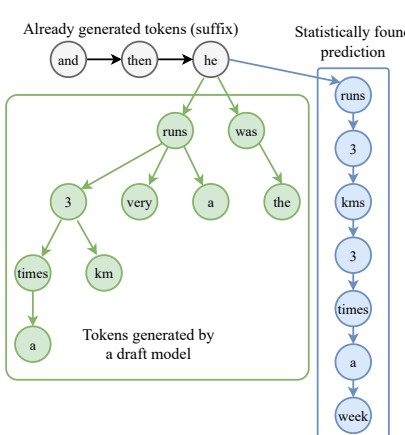

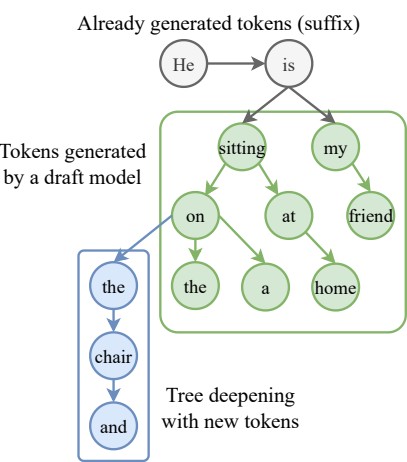

(2a) Appending a branch to the draft model tree. If both the branch and the tree have fixed structures, the resulting tree will also maintain a fixed structure. For maximum acceleration, the branch obtained from statistical search should be significantly deeper than the draft model tree.

(2b) An example of deepening the first branch of the draft model tree using statistical search. In this case, the generated tokens are "He" and "is" (the root token in the draft model corresponds to the last accepted token). First, the algorithm runs the draft model to generate the draft tree. We then attempt to extend the branch "sitting on" by performing a one-branch search using the tokens "He", "is", "sitting", "on".

Unlike purely model-based drafting, the wall-clock time of our drafting stage depends only on the size of the produced tree. Trie and suffix-array queries run on CPU and can proceed in parallel with the model-based speculator. Moreover, each sample in a batch searches independently, enabling per-sample multithreading. With multithreading, search latency is effectively determined by the final tree size. Because statistical search is substantially faster than draft-model calls, the overall drafting time is dominated by the latter.

**Draft Tree Widening.** We widen the draft model's proposal by *attaching a single retrieval-driven path* to the root of the draft tree. Suppose the draft model emits a fixed-shape tree of depth $d_{\text{draft}}$. In parallel, we build a deterministic path of depth $d_{\text{search}}$ (often $d_{\text{search}} \gg d_{\text{draft}}$) via search over the datastore and trie keyed by the current context suffix $S$. At each level, candidate continuations are ranked by datastore frequency; we append the top continuation and proceed.

This attachment preserves a *fixed per-sample tree shape*, enabling efficient batching, while exploiting: (i) the low marginal cost of CPU-side, parallelizable search, which supports a much deeper branch with negligible latency; and (ii) increased token diversity—search-derived tokens frequently differ from the draft model's proposals, expanding the candidate set and improving acceptance.

Empirically, adding multiple search branches increases verifier load without improving accepted length; therefore we attach exactly one deep branch (see Figure 2a).

**Draft Tree Deepening.** We also *deepen* the tree by using draft-model tokens to seed search. Concretely, we take a suffix of the generated sequence together with a prefix from a draft-model branch, then search for continuations of this combined sequence. We attach only a single resulting branch, as using multiple branches yielded no measurable gains in our experiments. The new branch is appended to the node where the search was initiated (illustrated in Figure 2b). Unlike widening, some appended tokens may go unverified if the draft prefix is later rejected by the base model. As before, we avoid merging intersecting tokens to preserve a fixed tree shape. This deepening is particularly effective at smaller batch sizes, where allocating a larger decoding tree is computationally feasible.

---

**Algorithm 1** READER

1: $y \leftarrow []; \quad t \leftarrow 0$
2: $\text{TRIEINSERT}(\mathcal{T}_{\text{short}}, x)$
3: **while** $t < T_{\max}$ **and** $\text{last}(y) \neq \text{EOS}$ **do**
4:    $c \leftarrow x \parallel y$                                                   {current decoding context}
5:    $\text{tree} \leftarrow \text{SPECULATOR}(c, d_{\text{draft}}, B_{\text{tokens}})$                        {building model-based draft tree}
6:    $\text{tree} \leftarrow \text{RETRIEVALAUGMENT}\big(c, \text{tree}, L_{\text{suffix}}, d_{\text{search}}, d_{\text{deep}}, \mathcal{T}_{\text{short}}, \mathcal{D}_{\text{long}}\big)$
7:    $\Delta \leftarrow \text{VERIFY}(c, \text{tree})$                                   {base model accepts a prefix}
8:    $y \leftarrow y \parallel \Delta$                                           {appending generated text}
9:    $\text{TRIEINSERT}(\mathcal{T}_{\text{short}}, \Delta)$                               {loading new data into the trie}
10:   **if** $t \bmod N = 0$ **then**
11:      $\text{KVCACHEREARRANGE}()$
12:   **end if**
13:   $t \leftarrow t + 1$
14: **end while**
15: **return** $y$

---

**Lossless Decoding.** We formalize that READER is *lossless* with respect to the base model's decoding policy. Intuitively, READER never alters the distribution — or, in the deterministic case, the exact identity — of the generated sequence; it only reduces the number of verifier calls. The proof of this can be found in Section D.

## 4.3 KV CACHE REARRANGEMENT

The methods above operate efficiently when KV cache movement is not the limiting factor — for 7-8B models this typically holds up to batch size 8. For larger batches, however, naive KV layout with per-prompt padding introduces avoidable memory traffic: when statistical search proposes a long accepted span, that span must be appended to the cache while other prompts are padded with zeros to match the new maximum length (see Figure 3a). In practice, most verification tokens come from the draft model, but intermittently the search path yields long acceptances, which increases the average acceptance length and, under padding, inflates the transient cache footprint.

This effect amplifies with batch size. Let $p$ be the probability that statistical search produces a long accepted span in a step, and $b$ the batch size. The probability that *at least one* prompt extends by a long span in that step is $1 - (1 - p)^b$, which rises rapidly with $b$. Thus, without care, larger batches induce disproportionately more padding and superfluous KV transfers.

We address this with a periodic *KV cache rearrangement* pass every $N$-th decoding step: for each prompt, we remove internal zero-padding and tightly concatenate non-zero entries, producing a compact, contiguous layout (Figure 3b). The hyperparameter $N$ trades compaction overhead against padding accumulation: too small $N$ adds unnecessary maintenance work; too large $N$ allows padding to grow. In our experiments we set $N = 25$, though the optimal value is hardware-dependent. Our strategy makes the KV cache overhead sublinear, compared to the sequence length.

Combining the widening and deepening strategies increases the total number of draft-tree nodes, yet with rearrangement the wall-clock time of draft generation remains comparable to a purely model-based speculator.

The scheme of READER method is presented in Algorithm 1. Function RETRIEVALAUGMENT consists of draft tree widening and optionally draft tree deepening. A comprehensive pseudocode for the READER method can be found in Section B.

## 5 EXPERIMENTS

### 5.1 SETUP

We evaluate READER across multiple LLMs, datasets, and batch sizes. Our external datastore is a suffix array built from the Magpie corpus (Xu et al., 2024), with no overlap with any evaluation set. Datastore and trie lookups run on CPU in parallel with the draft model; because these lookups finish well before the draft model call, they do not increase the end-to-end drafting time.

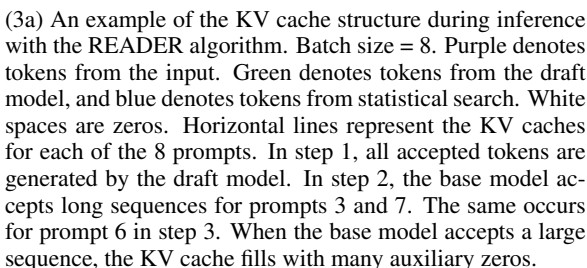

(3a) An example of the KV cache structure during inference with the READER algorithm. Batch size = 8. Purple denotes tokens from the input. Green denotes tokens from the draft model, and blue denotes tokens from statistical search. White spaces are zeros. Horizontal lines represent the KV caches for each of the 8 prompts. In step 1, all accepted tokens are generated by the draft model. In step 2, the base model accepts long sequences for prompts 3 and 7. The same occurs for prompt 6 in step 3. When the base model accepts a large sequence, the KV cache fills with many auxiliary zeros.

(3b) After rearranging, zeros may only remain at the end of the prompts. Total KV cache size is decreased.

| Model | Method | GSM | HumanEval | MT-Bench |
|---|---|---|---|---|
| | Autoregression | 1.00x | 1.00x | 1.00x |
| **Llama2-7B** | Lookahead | 1.65x | 2.03x | 1.72x |
| | Search + datastore (ours) | 2.91x | 3.27x | 2.84x |
| | PLD | 1.32x | 1.52x | 1.38x |
| | REST | 1.39x | 1.66x | 1.44x |
| | EAGLE | 2.71x | 3.16x | 2.58x |
| | EAGLE (opt. tree) | 3.02x | 3.58x | 2.87x |
| | READER (EAGLE backend) | **3.94x** | **4.30x** | **3.32x** |
| **Vicuna-7B** | PLD | 1.82x | 1.82x | 1.61x |
| | REST | 1.53x | 1.78x | 1.69x |
| | Medusa | 1.89x | 2.02x | 1.91x |
| | EAGLE | 2.61x | 3.30x | 2.52x |
| | READER (EAGLE backend) | **3.09x** | **4.24x** | **2.76x** |
| **Llama3.1-8B** | EAGLE-2 | 3.44x | 3.51x | 3.10x |
| | EAGLE-3 | 4.42x | 4.90x | 4.36x |
| | READER (EAGLE-3 backend) | **5.02x** | **6.13x** | **4.97x** |

Table 2: Speedup ratio vs. autoregressive decoding (temperature 0) on GSM, HumanEval, and MT-Bench for batch size 1. EAGLE (opt. tree) is an original EAGLE algorithm with tree structure optimized by our method.

Experiments are conducted on Llama2-7B (Touvron et al., 2023), Llama3.1-8B (Grattafiori et al., 2024) and Vicuna-7B (Chiang et al., 2023) using the GSM (Cobbe et al., 2021) dataset for mathematical reasoning, HumanEval (Chen et al., 2021) for code generation and MT-Bench (Zheng et al., 2023) for general prompts. We compare our method to training-free (Lookahead (Zhao et al., 2024), PLD (Saxena, 2023), REST (He et al., 2024)) and model-based (EAGLE (Li et al., 2024a;b; 2025), Medusa (Cai et al., 2024)) methods. Results are presented in Table 2 for batch size 1 and in Table 3 for larger batch sizes. Test results for sampling inference (temperature 1) are presented in Section E. We employ an optimized tree structure obtained via a search-based optimization algorithm, detailed in Section F. The resulting tree structures are also included in Section G.

We further evaluate READER on a RAG task using the hagrid dataset (Kamalloo et al., 2023), where the model must find the correct answer in a given context. Speculative decoding typically achieves high acceleration on such tasks, as the draft model can easily predict the continuation by copying some parts of the text from the prompt. However, model-based approaches are constrained by the number of draft model calls they can afford. In contrast, our method enables the generation of long continuations at almost no additional cost, resulting in a significantly higher average acceptance length and improved acceleration. Detailed results are provided in Table 4.

| Model | Method | GSM | | | HumanEval | | | MT-Bench |
|-------|--------|-----|-----|-----|-----------|-----|-----|----------|
| | | Batch size | | | Batch size | | | Batch size |
| | | 8 | 16 | 32 | 8 | 16 | 32 | 32 |
| | Autoregression | 1.00x | 1.00x | 1.00x | 1.00x | 1.00x | 1.00x | 1.00x |
| **Llama2-7B** | Lookahead | 1.61x | 1.53x | 1.38x | 1.94x | 1.56x | 1.40x | 1.32x |
| | Search + datastore (ours) | 2.66x | 2.71x | 2.31x | 3.13x | 2.70x | 2.25x | 1.98x |
| | EAGLE | 2.50x | 2.51x | 1.82x | 3.00x | 2.79x | 1.84x | 1.60x |
| | EAGLE (opt. tree) | 2.84x | 2.77x | 2.28x | 3.40x | 2.97x | 2.23x | 2.01x |
| | READER (EAGLE backend) | **3.63x** | **3.25x** | **2.66x** | **4.18x** | **3.56x** | **2.65x** | **2.29x** |
| **Vicuna-7B** | EAGLE | 2.43x | 2.35x | 2.10x | 3.13x | 2.65x | 2.01x | 1.73x |
| | READER (EAGLE backend) | **2.97x** | **2.89x** | **2.56x** | **3.95x** | **3.26x** | **2.37x** | **2.03x** |
| **Llama3.1-8B** | EAGLE-3 | 4.35x | 4.05x | 3.42x | 4.80x | 4.46x | 3.77x | 3.25x |
| | READER (EAGLE-3 backend) | **4.92x** | **4.43x** | **3.99x** | **5.92x** | **5.21x** | **4.34x** | **3.67x** |

Table 3: Speedup ratio vs. autoregressive decoding (temperature 0) on GSM, HumanEval, and MT-Bench for batch sizes 8, 16, and 32. EAGLE (opt. tree) is an original EAGLE algorithm with tree structure optimized by our method.

Table 4: Test results on the hagrid dataset for RAG tasks. All tests were run with a batch size 1 and Llama3.1-8B model.

| Method | Speedup | Avg. Acceptance Length |
|--------|---------|------------------------|
| Autoregression | 1.00x | 1.00 |
| EAGLE-3 | 5.31x | 6.7 |
| READER | **10.24x** | **14.03** |

## 5.2 ABLATION STUDY

We assess the contribution of each component of READER on GSM, HumanEval, and MT-Bench.

- *Tree Optimization.* Search-based adjustment of the draft tree improves throughput by ∼10% for batch sizes 8-16 and up to ∼23% at batch size 32.

- *Statistical Search Branch.* Adding one retrieval-driven branch yields the largest gain (∼20% on average), with the strongest effect at small batches where expansion is inexpensive.

- *Tree Deepening.* Seeding search from draft-model prefixes provides modest improvements (∼5% at batch size 8) and is enabled only for small-batch inference.

- *KV Cache Rearrangement.* Periodic compaction (every 25 steps for batch size 32, every 50 for size 16) lowers generation time by ∼7-8%.

## 6 CONCLUSION

This paper introduces READER, a lossless speculative decoding framework that recasts speculative decoding as a stochastic tree-construction problem and instantiates a retrieval-assisted drafter to realize this formulation in practice without additional training. Our complexity-theoretic analysis delineates the optimality frontier under bounded computation and memory, and we further present a memory-optimal KV cache serving strategy that guarantees amortized sublinear overhead in the batch dimension. READER preserves exact output equivalence while achieving up to 6.13x wall-clock speedup on single-prompt inference and up to 5.92x in batched settings, with more than 10x gains on RAG pipelines. By exploiting the empirical redundancy of natural language through a theoretically grounded draft-tree design, READER closes a key gap between the parallelism limits suggested by theory and practical LLM inference, pointing to a new standard for efficient deployment.

## 7 REPRODUCIBILITY STATEMENT

The comprehensive description with a pseudocode of READER can be found in Sections B and 4.

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

## A    USAGE OF LLMs

We used LLMs in our paper for improving readability and better formatting.

## B    READER PSEUDOCODE

---

**Algorithm 2** READER: Retrieval-Enhanced Drafting for Speculative Decoding

---

**Inputs**:
- $x$ — tokenized prompt
- $\mathcal{T}_{\text{short}}$ — short-term trie (prompt + self-generated history + optional external text)
- $\mathcal{D}_{\text{long}}$ — long-term datastore indexed by a suffix array

**Parameters**:
- $L_{\text{suffix}}$ — suffix length for search keys
- $d_{\text{draft}}$ — depth of draft-model tree
- $d_{\text{search}}$ — depth of the (single) retrieval branch for widening
- $d_{\text{deep}}$ — max depth of the (single) deepening branch per step
- $B_{\text{tokens}}$ — max token budget per step (fixed shape)
- $N$ — KV-cache rearrangement period
- EOS, $T_{\text{max}}$ — end conditions

**Black boxes**:
- SPECULATOR$(x, d_{\text{draft}}, B_{\text{tokens}})$ — proposes a fixed-shape draft tree
- VERIFY$(x, \text{tree})$ — base-model verification; returns accepted prefix $\Delta$

**Output**: completed sequence $y$

```
 1: y ← [];    t ← 0
 2: TRIEINSERT(𝒯_short, x)
 3: while t < T_max and last token of y ≠ EOS do
 4:     S ← RIGHTSUFFIX(x ∥ y, L_suffix)
 5:     tree ← SPECULATOR(x ∥ y, d_draft, B_tokens)
 6:     // Widen: attach one deep retrieval-driven branch at root
 7:     P_wide ← SEARCHPATH(S, d_search, 𝒯_short, 𝒟_long)
 8:     ATTACHATROOT(tree, P_wide)
 9:     // Deepen: seed search from a draft prefix, attach one branch
10:     u ← PICKDRAFTNODEFORDEEPENING(tree)
11:     S′ ← CONCAT(S, PREFIXFROMROOT(u))
12:     P_deep ← SEARCHPATH(S′, d_deep, 𝒯_short, 𝒟_long)
13:     ATTACHATNODE(tree, u, P_deep)
14:     // Verify and accept
15:     Δ ← VERIFY(x ∥ y, tree)                {accepts a contiguous prefix of some path}
16:     y ← y ∥ Δ;    t ← t + 1
17:     TRIEINSERT(𝒯_short, Δ)
18:     if t mod N = 0 then
19:        KVCACHEREARRANGE()
20:     end if
21: end while
22: return y
```

---

---

**Algorithm 3** SearchPath: CPU-side retrieval over trie + suffix array

---

**Input**: suffix $S$, depth $d$, structures $(\mathcal{T}_{\text{short}}, \mathcal{D}_{\text{long}})$
**Output**: path $P = (t_1, \ldots, t_k)$ with $k \leq d$
1: $P \leftarrow []; \quad C \leftarrow S$
2: **for** $i = 1$ **to** $d$ **do**
3: $\quad A \leftarrow \text{TRIENEXTTOKENS}(\mathcal{T}_{\text{short}}, C)$
4: $\quad B \leftarrow \text{SUFFIXARRAYNEXTTOKENS}(\mathcal{D}_{\text{long}}, C)$
5: $\quad L \leftarrow \text{RANKBYFREQUENCY}(A \cup B)$           {stable sort: higher freq first}
6: $\quad$ **if** $L = \emptyset$ **then**
7: $\quad\quad$ **break**
8: $\quad$ **end if**
9: $\quad t^\star \leftarrow L[1]$                                     {take top continuation}
10: $\quad P \leftarrow P \parallel t^\star; \quad C \leftarrow C \parallel t^\star$
11: **end for**
12: **return** $P$

---

**Algorithm 4** PickDraftNodeForDeepening (fixed-shape friendly)

---

**Input**: draft tree tree
**Output**: node $u$ for deepening
1: $u \leftarrow$ first node on the leftmost (max-prob) path at depth $\min(2, d_{\text{draft}})$
2: **return** $u$

---

**Algorithm 5** KVCacheRearrange (periodic compaction)

---

1: **for each** sample cache $\mathcal{KV}$ in batch (in parallel) **do**
2: $\quad \mathcal{KV} \leftarrow \text{REMOVEINTERNALZEROPADDING}(\mathcal{KV})$
3: $\quad \mathcal{KV} \leftarrow \text{TIGHTCONCATENATE}(\mathcal{KV})$
4: **end for**

---

**Algorithm 6** Verify (standard speculative verification sketch)

---

**Input**: context $c$, heterogeneous tree tree
**Output**: accepted contiguous token span $\Delta$
1: Expand base model along candidate paths in tree (batched by depth)
2: Compare base logits with drafted tokens; find longest prefix consistent with base
3: **return** that prefix as $\Delta$ (possibly length 0)

---

## C   PROOFS

### C.1   PROOF OF LEMMA 1

*Proof.* For each non-root node $i$, define the indicator $I_i = \mathbf{1}\{\text{node } i \text{ is accepted}\}$. Exactly those nodes on the verified path are accepted, so the total number of accepted tokens is

$$N = \sum_{i=1}^{t} I_i.$$

Taking expectations and using linearity,

$$\mathbb{E}[N] = \sum_{i=1}^{t} \mathbb{E}[I_i] = \sum_{i=1}^{t} \Pr(I_i = 1).$$

By definition of $\alpha_i$ as the cumulative probability of the node's prefix, $\Pr(I_i = 1) = \alpha_i$. Hence

$$\mathbb{E}[N] = \sum_{i=1}^{t} \alpha_i. \qquad \square$$

### C.2   PROOF OF LEMMA 2

*Proof.* Let the node set be $V = \{1, \ldots, t\}$. In a given speculative-drafting routine, nodes are produced by (possibly mixed) mechanisms:

(i) *Layerwise:* for each produced layer $l$ there is a subset $V_l \subseteq V$ of nodes generated by a single forward pass with cost $t_1^l$;

(ii) *Treewise:* there is a subset $V_{\mathrm{tw}} \subseteq V$ of nodes generated by a single forward pass with cost $t_2$;

(iii) *Nodewise:* there is a subset $V_{\mathrm{nw}} \subseteq V$ of nodes generated individually, where node $i$ takes time $t_3^i$.

These subsets form a disjoint partition of $V$ up to the natural indexing by layers: $V = \left(\bigsqcup_l V_l\right) \sqcup V_{\mathrm{tw}} \sqcup V_{\mathrm{nw}}$.

The *true* wall-clock drafting time is, by definition of the mechanisms above,

$$T_D^{\mathrm{true}} = \sum_l t_1^l + t_2 + \sum_{i \in V_{\mathrm{nw}}} t_3^i.$$

Define the *effective nodewise generation time* $t_i$ for each node $i \in V$ by

$$t_i = \begin{cases} \dfrac{t_1^l}{|V_l|}, & i \in V_l \text{ (layerwise)}, \\ \dfrac{t_2}{|V_{\mathrm{tw}}|}, & i \in V_{\mathrm{tw}} \text{ (treewise)}, \\ t_3^i, & i \in V_{\mathrm{nw}} \text{ (nodewise)}. \end{cases}$$

Then summing over all nodes yields

$$\sum_{i=1}^{t} t_i = \sum_l \sum_{i \in V_l} \frac{t_1^l}{|V_l|} + \sum_{i \in V_{\mathrm{tw}}} \frac{t_2}{|V_{\mathrm{tw}}|} + \sum_{i \in V_{\mathrm{nw}}} t_3^i = \sum_l t_1^l + t_2 + \sum_{i \in V_{\mathrm{nw}}} t_3^i = T_D^{\mathrm{true}}.$$

Hence the total drafting time equals the sum of the effective nodewise times, which proves

$$T_D\big(T(\alpha_1, \ldots, \alpha_t), t_1, \ldots, t_t\big) = \sum_{i=1}^{t} t_i.$$

This construction shows that type 1 (layerwise) and type 2 (treewise) nodes can be *accounted for* as type 3 (nodewise) nodes without changing the total time — simply distribute the cost of each group's forward pass uniformly over the nodes it produces. The argument trivially extends to multiple treewise groups $\{V_{\text{tw}}^{(g)}\}_g$ with costs $\{t_2^{(g)}\}_g$ by replacing $t_2/|V_{\text{tw}}|$ with $t_2^{(g)}/|V_{\text{tw}}^{(g)}|$ and summing over $g$. $\qquad\square$

The illustration of this lemma is shown in Figure 4.

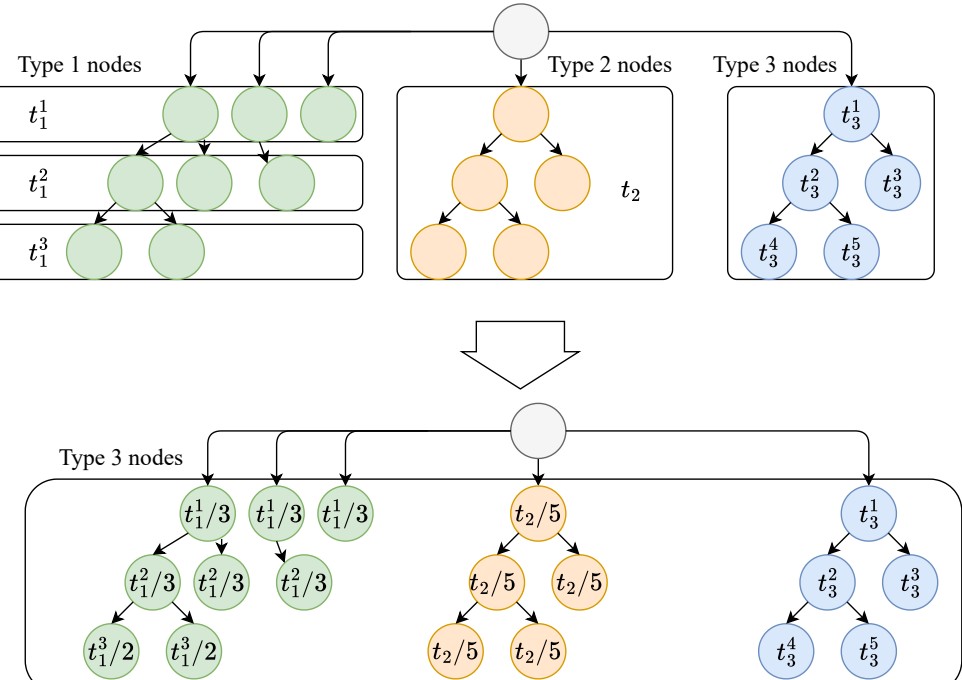

Figure 4: Converting type 1 and type 2 nodes into equivalent type 3 generation times.

## C.3 PROOF OF THEOREM 1

*Proof.* Fix a rooted decoding tree $T$ with $|T|$ nodes, acceptance frequencies $\alpha_1, \ldots, \alpha_t$ (where $\alpha_i = \Pr(\text{node } i \text{ is accepted})$), and effective nodewise generation times $t_1, \ldots, t_{|T|}$. Consider one speculative step that (i) drafts the tree $T$ and (ii) verifies it.

Let $A_T$ be the random number of tokens accepted in this step, and let $C_T$ be the random time spent in this step (drafting + verification). Running the same procedure repeatedly with the same $T$, the long-run throughput (tokens per second) is

$$\text{throughput}(T) \;=\; \lim_{N \to \infty} \frac{\sum_{n=1}^{N} A_T^{(n)}}{\sum_{n=1}^{N} C_T^{(n)}} \;=\; \frac{\mathbb{E}[A_T]}{\mathbb{E}[C_T]},$$

where the equality follows from the law of large numbers.

By Lemma 1, the expected number of accepted tokens in a single verification of $T$ is

$$\mathbb{E}[A_T] \;=\; E(T) \;=\; \sum_{i=0}^{|T|} \alpha_i.$$

Decompose the step time into drafting and verification:

$$\mathbb{E}[C_T] \;=\; T_D(T) \;+\; T_V(T).$$

By Lemma 2, the drafting time aggregates nodewise as

$$T_D(T) \;=\; \sum_{i=1}^{|T|} t_i,$$

so that

$$\mathbb{E}[C_T] \;=\; \sum_{i=1}^{|T|} t_i \;+\; T_V(T).$$

Therefore the throughput achieved by $T$ is

$$\frac{\mathbb{E}[A_T]}{\mathbb{E}[C_T]} \;=\; \frac{\sum_{i=0}^{|T|} \alpha_i}{\sum_{i=1}^{|T|} t_i + T_V(T)} \;=\; \left( \sum_{i=0}^{|T|} \alpha_i \right) \cdot \left( \sum_{i=1}^{|T|} t_i + T_V(T) \right)^{-1}.$$

Maximizing throughput over all rooted trees $T$ is thus equivalent to solving

$$T^* \in \underset{T \text{ — rooted tree}}{\arg\max} \left( \sum_{i=0}^{|T|} \alpha_i \right) \cdot \left( \sum_{i=1}^{|T|} t_i + T_V(T) \right)^{-1},$$

which is exactly the statement of Theorem 1. $\qquad\square$

# D  LOSSLESS DECODING OF READER

Let $P_\theta(\cdot \mid x_{<t})$ denote the base model's conditional distribution at step $t$. Let $\pi$ be the target decoding policy (e.g., greedy/top-1 with fixed tie-break; or sampling with temperature and top-$p$/top-$k$ filtering). Let $X_{1:T}$ be the sequence produced by running the *base model alone* with policy $\pi$. Let $\widehat{X}_{1:T}$ be the sequence produced by READER.

We say that READER is *lossless* if and only if:

- for deterministic $\pi$ (e.g., greedy), pathwise equality holds: $\widehat{X}_{1:T} = X_{1:T}$;

- for randomized $\pi$ (e.g., temperature + top-$p$/$k$ sampling), the laws agree: $\widehat{X}_{1:T} \stackrel{d}{=} X_{1:T}$.

READER builds speculative draft trees using any mixture of model proposals and search-derived continuations (trie/suffix-array/datastore). Correctness requires only the following invariants:

1. The *verifier* is the same base model $P_\theta$ using the same temperature, filtering, and tie-breaking as $\pi$.

2. A token is *committed* only after explicit verification against $P_\theta(\cdot \mid \text{current prefix})$. Multi-token acceptance means committing the longest prefix that matches the verifier step-by-step.

3. Upon the first mismatch, all speculative tokens beyond that point are discarded, and the next token is obtained by applying $\pi$ to $P_\theta$ at the current prefix.

4. For randomized $\pi$, READER uses the same underlying randomness as the base run (formalized via coupling below).

**Deterministic decoding (greedy/top-1).**  Assume $\pi(P_\theta(\cdot \mid x_{<t})) = \arg\max_y P_\theta(y \mid x_{<t})$ with a fixed tie-break rule.

**Theorem (Deterministic losslessness).** *Under the verification invariants, $\widehat{X}_{1:T} = X_{1:T}$.*

*Proof.* We prove by induction on $t$ that after committing at step $t$, $\widehat{X}_t = X_t$. For $t = 1$, READER either accepts a proposed token equal to $\arg\max P_\theta(\cdot \mid \emptyset)$ or rejects and falls back to that argmax; in both cases $\widehat{X}_1 = X_1$. Assume $\widehat{X}_{<t} = X_{<t}$. By the acceptance rule, READER accepts a proposed $y_t$ only if $y_t = \arg\max_y P_\theta(y \mid X_{<t})$; otherwise it rejects and commits exactly that argmax. Hence $\widehat{X}_t = \arg\max_y P_\theta(y \mid X_{<t}) = X_t$. Therefore $\widehat{X}_{1:T} = X_{1:T}$. $\square$

**Randomized decoding (sampling).**  Let $\pi$ be any randomized policy implementable as a measurable map

$$F : \left(\Delta^{|\mathcal{V}|-1}, [0,1]\right) \to \mathcal{V} \quad \text{such that} \quad X_t = F\left(P_\theta(\cdot \mid X_{<t}), U_t\right),$$

for i.i.d. uniforms $U_t \sim \text{Unif}[0,1]$; this covers temperature scaling, top-$p$/top-$k$ filtering, and inverse-CDF sampling (with deterministic tie-breaking on measure-zero boundaries).

**Theorem (Randomized losslessness).** *Couple READER and the base run with the* same *i.i.d. uniforms $(U_t)_{t \geq 1}$. Under the verification invariants, $\widehat{X}_{1:T}$ and $X_{1:T}$ are equal almost surely, hence $\widehat{X}_{1:T} \stackrel{d}{=} X_{1:T}$.*

*Proof.* Induct on $t$. Assume $\widehat{X}_{<t} = X_{<t}$. Let $x_t^\star := F(P_\theta(\cdot \mid X_{<t}), U_t)$ be the token the base policy would emit. READER verifies its speculative draft at prefix $X_{<t}$. If $x_t^\star$ appears among the verified candidates at step $t$, READER commits it. If not, READER discards the speculative step and *falls back* to committing $F(P_\theta(\cdot \mid X_{<t}), U_t) = x_t^\star$. Thus in all cases $\widehat{X}_t = x_t^\star = X_t$ almost surely. Therefore the entire sequences coincide almost surely. $\square$

Under the stated invariants, READER commits exactly the tokens that the base model would produce under $\pi$ (pathwise for deterministic $\pi$; in distribution for randomized $\pi$). Hence, READER is *lossless*.

# E    TEST RESULTS WITH SAMPLING

The test results for speculative decoding with sampling with temperature 1 are presented in Table 5 for batch size 1 and in Table 6 for larger batch sizes.

| Model | Method | GSM | HumanEval | MT-Bench |
|---|---|---|---|---|
| | Autoregression | 1.00x | 1.00x | 1.00x |
| **Llama2-7B** | Lookahead | 1.40x | 1.71x | 1.42x |
| | Search + datastore (ours) | 2.38x | 2.71x | 2.34x |
| | PLD | 1.11x | 1.25x | 1.14x |
| | REST | 1.16x | 1.41x | 1.19x |
| | EAGLE | 2.30x | 2.59x | 2.09x |
| | EAGLE (opt. tree) | 2.45x | 2.87x | 2.35x |
| | READER (EAGLE backend) | **3.19x** | **3.56x** | **2.69x** |
| **Vicuna-7B** | PLD | 1.53x | 1.52x | 1.32x |
| | REST | 1.30x | 1.52x | 1.43x |
| | Medusa | 1.61x | 1.66x | 1.61x |
| | EAGLE | 2.22x | 2.77x | 2.09x |
| | READER (EAGLE backend) | **2.50x** | **3.50x** | **2.30x** |
| **Llama3.1-8B** | EAGLE-2 | 2.82x | 2.89x | 2.48x |
| | EAGLE-3 | 3.37x | 4.15x | 3.02x |
| | READER (EAGLE-3 backend) | **3.96x** | **5.17x** | **3.82x** |

Table 5: Speedup ratio vs. decoding with sampling (temperature 1) on GSM, HumanEval, and MT-Bench for batch size 1.

| Model | Method | GSM | | | HumanEval | | | MT-Bench |
|---|---|---|---|---|---|---|---|---|
| | | Batch size | | | Batch size | | | Batch size |
| | | 8 | 16 | 32 | 8 | 16 | 32 | 32 |
| | Autoregression | 1.00x | 1.00x | 1.00x | 1.00x | 1.00x | 1.00x | 1.00x |
| **Llama2-7B** | Lookahead | 1.36x | 1.29x | 1.11x | 1.59x | 1.30x | 1.18x | 1.09x |
| | Search + datastore (ours) | 2.25x | 2.18x | 1.96x | 2.62x | 2.24x | 1.84x | 1.63x |
| | EAGLE | 2.07x | 2.15x | 1.54x | 2.48x | 2.36x | 1.52x | 1.38x |
| | EAGLE (opt. tree) | 2.32x | 2.33x | 1.94x | 2.86x | 2.54x | 1.90x | 1.70x |
| | READER (EAGLE backend) | **2.99x** | **2.64x** | **2.17x** | **3.35x** | **2.98x** | **2.19x** | **1.88x** |
| **Vicuna-7B** | EAGLE | 1.98x | 1.92x | 1.77x | 2.61x | 2.26x | 1.66x | 1.52x |
| | READER (EAGLE backend) | **2.44x** | **2.46x** | **2.13x** | **3.28x** | **2.64x** | **1.99x** | **1.77x** |
| **Llama3.1-8B** | EAGLE-3 | 3.29x | 3.21x | 2.83x | 4.04x | 3.81x | 3.19x | 2.34x |
| | READER (EAGLE-3 backend) | **3.91x** | **3.71x** | **3.28x** | **4.76x** | **4.28x** | **3.57x** | **2.80x** |

Table 6: Speedup ratio vs. decoding with sampling (temperature 1) on GSM, HumanEval, and MT-Bench for batch sizes 8, 16, and 32.

## F  CHOOSING TREE STRUCTURE

Fixed-size draft trees offer simplicity, but the *throughput-optimal* configuration depends on the base model, batch size, and hardware characteristics (e.g., KV-cache bandwidth, memory hierarchy, and compute occupancy). These factors interact nonlinearly, making a closed-form derivation of the optimal tree size and shape intractable.

We identify tree configurations *empirically* while guarding against dataset-specific bias. For example, code-generation workloads can favor deeper trees than arithmetic reasoning, but such preferences do not always transfer. We therefore calibrate using a general-purpose benchmark (MT-Bench (Zheng et al., 2023)) to obtain settings that generalize across tasks.

We begin from an overprovisioned seed tree $T_0$. Running the drafter/verification loop, we estimate for each node $v$ its acceptance probability $\alpha(v)$, defined as the probability that $v$ contributes to an ultimately accepted continuation. Empirically and by construction of speculative verification, descendants have acceptance at most that of their ancestors, i.e.,

$$\alpha(\text{child}) \leq \alpha(\text{parent}).$$

We exploit this monotonicity by iteratively pruning the $n$ lowest-acceptance *leaves* to obtain $T_n$. This preserves connectivity and ensures $T_n$ remains a valid prefix subtree of $T_0$ at every iteration.

Our goal is to select the pruning level $n_\star$ that maximizes end-to-end throughput (tokens/sec), accounting for drafting, verification, and KV/cache effects:

$$n_\star \in \arg\max_n \text{ throughput}(T_n).$$

We treat throughput as a black-box objective and apply Bayesian optimization (Gardner et al., 2014), where each benchmarked tree $T_n$ yields a (noisy) observation of the target metric. In practice, the response curve in $n$ is close to unimodal; when wall-clock budget is tight, a golden-section search provides a lightweight alternative with comparable solutions.

A schematic of one pruning iteration—estimating per-node acceptance and removing the lowest-acceptance leaves—is shown in Figure 5.

## G  DECODING TREE STRUCTURES

This subsection presents the tree structures used for the Llama2-7B test results in the paper.

For a batch size of 8, the tree structure is shown in Figure 6, which incorporates both the statistical search branch and tree deepening methods.

For batch sizes 16 and 32, the corresponding tree structures are shown in Figures 7 and 8, respectively. For these batch sizes, only the statistical search branch is applied in the decoding trees.

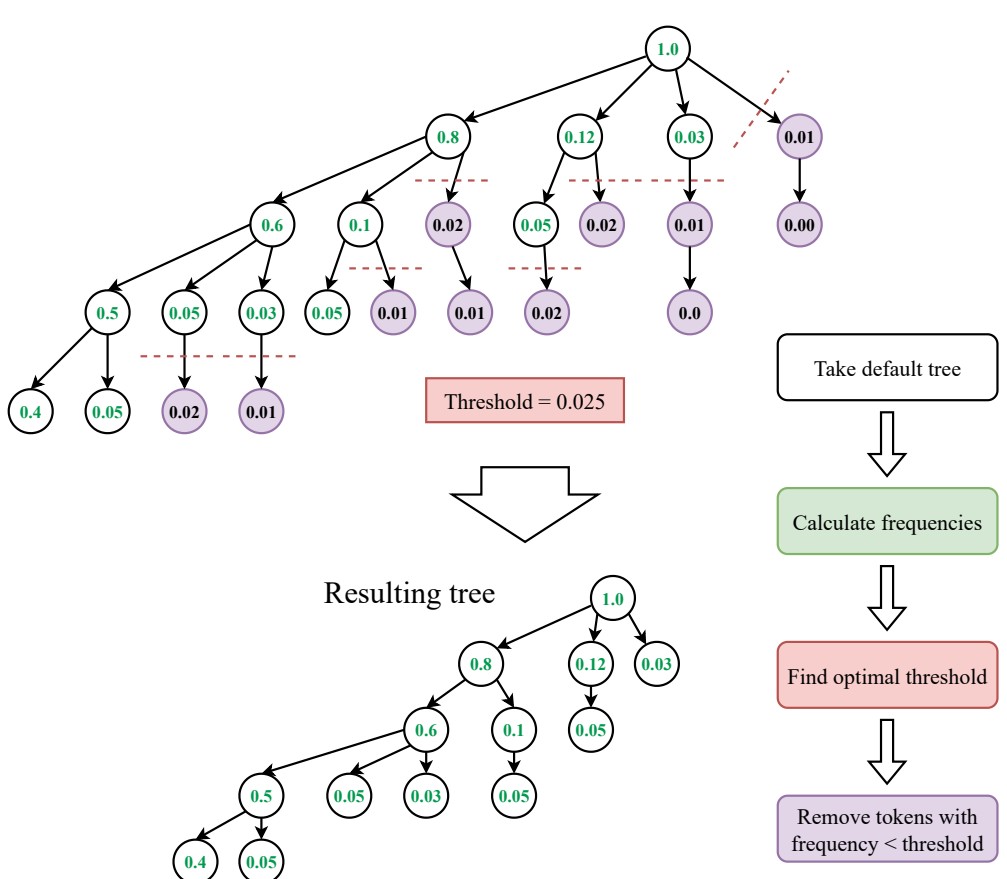

Figure 5: An example of the tree structure determination workflow. The process begins with a benchmark on an excessively large tree to obtain node acceptance rates. Subsequently, subtrees with acceptance probabilities below a threshold are removed.

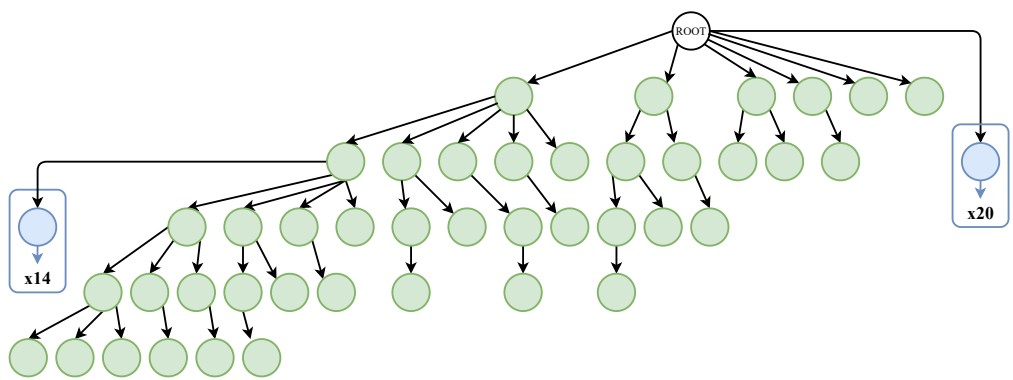

Figure 6: Decoding tree structure for batch size 8.

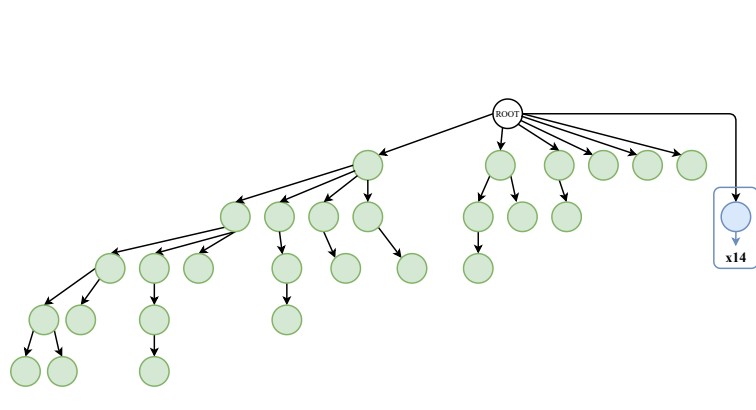

Figure 7: Decoding tree structure for batch size 16.

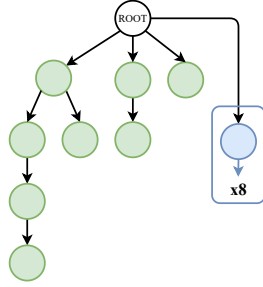

Figure 8: Decoding tree structure for batch size 32.

