# OpenReview forum: "READER: Retrieval-Assisted Drafter for Efficient LLM Inference"
_ICLR.cc/2026/Conference — Submitted to ICLR 2026_

### Official Review · Reviewer_CBpS · 2025-10-14

**Soundness:** 3
**Presentation:** 2
**Contribution:** 3
**Rating:** 4
**Confidence:** 5

**Summary:**

The paper proposes READER, a lossless speculative decoding method that augments an existing draft model with a statistical search branch drawn from self-repetitions and a datastore, thereby deepening the decoding tree and increasing the average acceptance length.

It also introduces a KV-cache rearrangement technique to mitigate memory and bandwidth bottlenecks in large-batch settings. In experiments, READER achieves speedups up to ~4–5× over AR decoding.

**Strengths:**

1. The combination of retrieval-based speculative decoding with model-based speculative decoding is quite interesting and novel, in my opionion.
2. The speedup number is quite impressive.

**Weaknesses:**

1. > bypasses the training of the auxiliary draft model

This is a bit misleading as the method still needs a trained draft model.

2. I feel Section 3 is also part of the methodology and maybe Section 4 should be renamed to improve clarity.

3. Section 4 is a bit confusing on its own. Most of the algorithm details are in appendix. It will be helpful if a more concise version pseudo-algorithm on the overall method is included in Section 4.

4. The evaluation is a bit too short (only one and a half page) and not comprehensive enough. First, why are most of the baseline results for MT-Bench omitted? Also, the ablation study should be accompanied with some tables or figures. For example, what are the raw values for the percentage improvement with tree optimization, statistical search branch, tree deepening, and KV cache rearrangement? What are the experiment setting (like GPUs used and models)? How is the accept length affected? In my opinion, the evaluation section is not well-written and is the major weakness of this paper.

5. In general, I feel the paper writing needs a lot of improvement. It should focus on highlighting the most important contributions (e.g. retrieval-augmented tree) clear, instead of explaining existing approach, like in Section 3 and 4. As a result of the lengthy methodology section, there is not enough room for evaluation, which significantly weakens the paper's claim.

6. Will the code be open-sourced?

**Questions:**

See the weaknesses above.

---

> ### Author Response · Authors · 2025-11-13
> **Response**
>
> We thank the reviewer for the time, thoughtful assessment and for the helpful suggestions for improvement. We also appreciate the positive remarks regarding the novelty of our approach and the strength of our empirical speedups. Below we address the concerns raised and describe the revisions we made.
>
> 1. We agree that the original sentence may lead to misunderstanding. We have rewritten it in the abstract as follows: "…speculative decoding framework that reuses an existing trained draft model as its backbone and requires no additional retraining…". We believe this version provides clearer insight into the structure of the algorithm.
> 2. To improve clarity, we renamed Section 4, as it relates only to our algorithm. Section 3 contains only the theoretical foundations on which our method is built. To further improve clarity, we added a description of the paper’s organization in the introduction. We also included a brief pseudocode sketch of our algorithm into the main text, while the complete pseudocode with auxiliary functions remains in the appendix. The updated introductory text referring to Sections 3-4 is as follows: "...Section 3 provides a comprehensive theoretical analysis of speculative decoding with a heterogeneous tree structure, establishing the foundations for our method. Section 4 presents the READER algorithm along with a brief pseudocode sketch...".
> 3. Discussed in 2.
> 4. We agree with this statement. To address it, we conducted comprehensive experiments on additional baseline methods, including REST, PLD, EAGLE-2, and Medusa. We also split the experiments into two tables: one for batch size 1 and another for larger batch sizes. The evaluation section has been substantially expanded. We performed experiments on MT-Bench dataset for all models. See Tables 2 and 3, as well as Tables 5 and 6, for greedy decoding and sampling respectively. The models used are stated in the "Model" column in each table. Regarding the ablation study, we report the average acceleration achieved across all batch sizes for GSM and HumanEval tasks.
> 5. The idea was to provide a comprehensive theoretical background that supports our method and can also be useful for other research on speculative decoding. Section 3.3 presents a novel result, and we consider Sections 3.1–3.2 to be the necessary formalism for our derivations. Section 3.2 is also essential for understanding why KV cache rearrangement is required. As mentioned above, we have largely expanded our experiments, and we hope that, with these additions, our method sounds stronger.
> 6. The release of the source code is planned.
>
> **Conclusion:**
>
> We hope that our revisions and expanded results address the reviewer's concerns and clarify the significance of our contribution. We appreciate the feedback, which has helped us improve the clarity of the paper. We kindly ask the reviewer to reconsider the evaluation in light of the newly provided results. We are glad to provide further clarification if the reviewer has additional questions.

---

### Official Review · Reviewer_xkaY · 2025-10-30

**Soundness:** 3
**Presentation:** 4
**Contribution:** 3
**Rating:** 6
**Confidence:** 3

**Summary:**

This paper introduces READER, a novel, training-free framework for accelerating Large Language Model inference via speculative decoding. The core contribution is the design of a heterogeneous draft tree that combines a standard, model-based speculator with a retrieval-assisted drafter. This retrieval component leverages both a short-term trie of the current context and a long-term, pre-built datastore (indexed by a suffix array) to generate high-probability, repetitive text sequences. The paper provides a theoretical analysis that frames speculative decoding as a tree-construction optimization problem and introduces practical system optimizations like a periodic KV cache rearrangement to maintain efficiency at large batch sizes. Experiments show that READER achieves significant wall-clock speedups (up to 6.13x) over standard autoregressive decoding and consistently outperforms strong speculative decoding baselines, with particularly pronounced gains on RAG tasks.

**Strengths:**

1. The hybrid approach of combining a learned draft model with a training-free, retrieval-based drafter is a novel and highly effective idea. This allows the system to leverage the best of both worlds: the draft model's ability to generate novel text and the retrieval system's efficiency in handling common, repetitive sequences.

2. The paper is well-grounded in theory, formalizing the speculative decoding process as a throughput optimization problem over a heterogeneous tree. This provides a principled foundation for the method and guides the design of the draft tree structure, elevating the work beyond a simple engineering heuristic.

3. The empirical results are strong and compelling. The method demonstrates significant and consistent speedups across multiple models and benchmarks. The over 10x speedup on RAG tasks is particularly impressive and highlights a key domain where this approach offers a transformative performance improvement.

4. The work is systems-aware, directly addressing the practical bottleneck of KV cache management at large batch sizes. The proposed periodic rearrangement is a simple yet effective solution that demonstrates a thoughtful approach to real-world deployment challenges.

**Weaknesses:**

1. The primary concern is that the method's impressive performance may be heavily skewed towards tasks with high textual repetition, which is the ideal scenario for a retrieval-based drafter. The remarkable >10x speedup on RAG, where the model copies extensively from the context, and the strong performance on code generation are clear evidence of this. However, the benefits might be substantially lower for more open-ended, creative, or complex reasoning tasks that require generating novel text with low repetition. This raises questions about the generality of the speedups and whether the method's effectiveness is confined to specific, repetition-heavy domains.

2. The proposed READER system introduces considerable complexity compared to purely model-based speculative decoding methods. It requires implementing and maintaining a short-term trie, building and loading a large external datastore with a suffix array, and managing parallel CPU-based search queries alongside GPU-based model inference. While the results are faster, this added engineering and maintenance overhead might be a significant barrier to adoption for some use cases.

3. The performance of the retrieval component is inherently dependent on the quality, size, and domain-relevance of the external datastore. While the paper uses a general-purpose dataset, deploying READER for a specialized domain would likely require curating a new, domain-specific datastore. This introduces a new dependency and potential for performance degradation if the datastore is poorly matched to the inference task, a limitation not present in self-contained, model-only approaches.

**Questions:**

1. Could you discuss the performance of READER on tasks characterized by low textual repetition, such as creative writing or abstract summarization? How much does the speedup diminish when the retrieval component finds few or no matches in the context or datastore?

2. How sensitive is READER's performance to the content and domain of the long-term datastore? For instance, how would the system perform on a medical QA task if the datastore is built from a general web corpus like the one used in the paper?

---

> ### Author Response · Authors · 2025-11-13
> **Response**
>
> We thank the reviewer for the positive evaluation of our work and for the thoughtful comments. Below we address the questions raised in the review.
>
> **General comments:**
>
> We expanded the experimental section with additional evaluations in Tables 2, 3 for temperature 0 and in Tables 5, 6 for temperature 1. We included results on MT-Bench, which consists of many tasks like QA, reasoning, understanding and other. This demonstrates the consistent performance of our method in real-world scenarios.
>
> **Questions:**
> 1. It is indeed correct that our method provides the most significant speedup on tasks with a high ratio of self-repetitions. Moreover, we analyze the maximum possible theoretical speedup achievable by a purely statistics-based method. This analysis is provided in Section 4.1 and in Table 1. However, compared to other purely statistical methods (e.g., REST, Prompt Lookup Decoding), our method still includes a model-based drafter. In fact, these approaches complement each other on different tasks.
>    In corner cases where there are no self-repetitions in the text (which is unrealistic in real data), the speedup converges to the speedup of the method used as the backbone. This speedup remains high for backbones like EAGLE-3 (up to 6x). On the other hand, if the answer to the prompt can be fully found within the prompt itself, the acceptance rate is limited only by the depth of the search-based branch. For example, for small batch sizes we use a depth of up to 20 tokens (see Figure 6).
> 2. Ideally, the data domain should match that of the inference stage. Due to this reason, we used the Magpie dataset, which covers a wide range of topics and includes coding tasks, math, and natural language. For instance, if the primary use case involves coding tasks, using a datastore consisting of code samples would substantially improve performance. However, outliers are still handled by the draft model, ensuring that overall performance does not degrade.
>
> **Conclusion:**
>
> We hope the detailed responses and added experiments address the concerns raised and better reflect the strengths of READER. We appreciate the reviewer's thoughtful evaluation.

---

> > ### Comment · Reviewer_xkaY · 2025-11-24
> >
> > Thank you for your response. I do maintain the concerns regarding the data domain and the requirement for self-repetitions, as well as the original strengths listed above. Hence, I will maintain the rating of 6.

---

### Official Review · Reviewer_GzNr · 2025-10-31

**Soundness:** 2
**Presentation:** 1
**Contribution:** 2
**Rating:** 2
**Confidence:** 4

**Summary:**

This paper presents READER (Retrieval-Assisted Drafter for Efficient LLM Inference), which accelerates speculative decoding by constructing a heterogeneous draft tree that blends search- (retrieval-) derived tokens with tokens proposed by a draft model.

**Strengths:**

1. The paper presents an investigation to combine both retrieval-based and drafting-based speculative decoding. The exploration efforts should be encouraged.
2. The authors conduct comprehensive experiments on a wide range of text generation benchmarks. The models include Llama2-7B, Vicuna-7B, and Llama3.1-8B.
3. It is worth noting that the authors consider the KV cache management in the speculative decoding paradigm and conduct experiments with batched settings, which are valuable in practice.

**Weaknesses:**

1. **The writing is poor and the demonstrations are confusing**. This manuscript is poor in writing, and readers may find it difficult to grasp the main contributions of the designed methodology. Detailed errors and problems are noted in the questions 1-5 below. I strongly recommend that the authors polish this manuscript further for the next submission.
2. **Lack of detail in the methodology**. I understand that READER aims to accelerate speculative decoding by constructing drafts augmented with search-based tokens. However, the writing of the methodology part is confusing and lacks many technical details. I could not effectively understand any specifics of the designed method. Particularly, your method is based on the Eagle series, right? Then, how do you construct the extra context or retrieval corpus for searching? What is the size (of tokens/documents) of your corpus? In Lines 308-309, you mentioned that "we augment the system with a large auxiliary datastore comprising many responses, ideally produced by the base model". How many responses are in this datastore? These details are missing.
3. **Lack of comprehensive experiments**. This manuscript only includes the experimental results of Tables 2 and 3. In Table 2, more baselines are required, such as REST for LLaMA-2-7B, Eagle-2, and other retrieval-based baselines, such as PLD and token recycling.



[1] Prompt Lookup Decoding. Apoorv Saxena. 2023. github.com/apoorvumang/prompt-lookup-decoding.

[2] Turning Trash into Treasure: Accelerating Inference of Large Language Models with Token Recycling. Luo et al. ACL 2025.

**Questions:**

Most of my primary concerns are outlined in the weaknesses section above. Here are additional minor concerns:

1. There exist multiple misuses of \citet and \citep in this manuscript. You could find errors in Lines 43, 143, and 440.
2. Too many grammar errors in this manuscript that severely impact the reading. For example, in Line 4-58, what is the meaning of "the cost of this step-by-step process scales poorly"?; In Line 51, "A promising line of work seeks to reduce this sequential bottleneck is speculative decoding"; In Line 163-164, the word "iff" should be fixed to "if".
3. The alternate use of "the base model" and "the main model" that refers to the target LM (or the model that conducts verification) may confuse readers.
4. Section 3.1, which introduces the background of speculative decoding, is poor in writing and does not clearly demonstrate the basics of the paradigm. For instance, what is the detailed process of "drafting-then-verify"? How does speculative decoding guarantee the lossless quality of the outputs?
5. Section 3.2 recaps the basics of tree decoding. However, it ignores the most important implementation in this technique, which is the tree attention masks in the verification process.

---

> ### Author Response · Authors · 2025-11-13
> **Response to the review**
>
> We thank the reviewer for the time, efforts and valuable feedback. We address all points raised below.
>
> **Questions:**
> 1. We thank the reviewer for noting this inconsistency. We have corrected all misuses of `\citep` and `\cite` throughout the paper.
> 2. We believe that sentence "As model sizes and context lengths grow, the cost of this step-by-step process scales poorly" is grammatically correct, however, we rewrote it to make the intended meaning more intuitive. The word "iff" was intentionally written as the standard mathematical abbreviation for "if and only it". We replaced it with a full phrase to avoid confusion. The sentence "A promising line of work seeks to reduce this sequential bottleneck is speculative decoding” has been corrected by adding the missing "that". We believe that the current version is now grammatically correct.
> 3. We now consistently use the term "base model." The usage of other terms has been removed.
> 4. Drafting-then-verify is a scheme used in most speculative decoding methods and is explained in a seminal paper (Leviathan et al., 2023), which is cited in line 052. It is also briefly described in our paper in lines 137-142. Regarding the lossless quality of the outputs, our approach is similar to other speculative decoding methods and is guaranteed by the verification stage. A formal proof for our method can be found in Section D.
> 5. Tree decoding is a commonly used technique in speculative decoding. We cite the papers (Miao et al., 2024; Sun et al., 2023) that introduced this approach. A short explanation is provided in lines 160–164. We also added a brief explanation of why this mask is needed at the verification stage.
>
> **Weaknesses:**
> 1. Discussed above.
> 2. In fact, our method is not based on the EAGLE family. We clarify that READER is a heterogeneous drafter that can be built on top of any speculator. We instantiate it with EAGLE/EAGLE-3 backends in our experiments. However, the use of any other method (e.g. Medusa) is possible. Regarding the retrieval corpus, the specific text used is the Magpie dataset, which is open-source. We state it in lines 429–432 and 323, and the dataset itself is cited.
> 3. We thank the reviewer for proposing additional methods for comparison. We compared our method with the most efficient speculative decoding approach available at the time of writing (namely, EAGLE-3). We have added extensive experiments to Tables 2 and 3 for greedy decoding and to Tables 5 and 6 for sampling.
>
> **General comment:**
>
> We improved readability by moving brief pseudocode to the main text and including paper organization paragraph into the introduction section.
>
>
> **Conclusion:**
>
> We believe the responses above address your concerns regarding writing clarity and comparisons to other methods. While presentation has been refined, the performance of our method is the central strength of the work. We hope the revised paper now meets the standards you outlined, and we kindly ask you to reconsider your evaluation with an emphasis on the technical contributions and empirical performance of READER.

---

### Meta-Review · Area_Chair_ffA8 · 2026-01-12

**Summary:**

The paper proposes a hybrid drafter for speculative decoding (SD), READER, which combines model-based drafter and retrieval-based drafter. The retrieval-based drafts are built through constructing a suffix array using the prompt and generation history. The paper empirically shows that the method provides higher speedup compared to the model-based drafting baseline.

During the discussion phase, reviewer raised concerns regarding the clarity of the presentation of the paper, and thoroughness of the experimental demonstration of the method regarding benchmark tasks and comparison to existing algorithms.

I agree with both raised concerns. While the paper partly addressed them in the rebuttal, I believe the following still stands: (1) it is unclear how the optimal tree principle in Section 3 is applied to the READER algorithm. It is briefly discussed in Section F, but the part about how to estimate the frequencies of the nodes and how Bayesian optimization is performed is not discussed in detail. I believe these are important aspects of the paper and should be discussed in main paper; (2) the paper missed an important existing baseline of SAM-decoding [Wu et al, 2024], which constructs suffix automation to enhance model-based drafter. The work is highly relevant and should be included as a baseline for the paper.

Hence I would recommend a rejection for the paper. And paper would benefit from addressing the above concerns and other concerns from the reviewers.

[Wu et al, 2024] SAM Decoding: Speculative Decoding via Suffix Automaton
Yuxuan Hu, Ke Wang, Xiaokang Zhang, Fanjin Zhang, Cuiping Li, Hong Chen, Jing Zhang

**Reviewer Concerns:**

See above discussion.

**Reviewer Scores:**

I wouldn't expect the reviewer scores to change.

---

### Decision · Program_Chairs · 2026-01-26

Reject